behaviour/ecology/evolution

despotic aggression, painted bunting, moulting, non-territorial aggression, multi-level selection

**Author for correspondence:**
Vanya G. Rohwer
e-mail: vgr7@cornell.edu

# Despotic aggression in pre-moulting painted buntings

Vanya G. Rohwer[1], Sievert Rohwer[2] and John C. Wingfield[3]

[1]Cornell University Museum of Vertebrates, Ithaca, NY 14850, USA
[2]Burke Museum of Natural History and Culture and Department of Biology, University of Washington, Seattle, WA 98195, USA
[3]Department of Neurobiology, Physiology and Behavior, University of California, One Shields Avenue, Davis, CA 95616, USA

VGR, 0000-0002-2505-3761

Aggression in territorial social systems is easy to interpret because the benefits of territorial defence mostly accrue to the territorial holder. However, in non-territorial systems, high aggression seems puzzling and raises intriguing evolutionary questions. We describe extreme rates of despotism between age classes in a passerine bird, the painted bunting (*Passerina ciris*), during the pre-moulting period. Aggressive encounters were not associated with aggressors gaining immediate access to resources. Instead, conspecifics, and even other species, were pursued as though being harassed; this aggression generated an ideal despotic habitat distribution such that densities of adult males were higher in high-quality sites. Aggression was not a by-product of elevated testosterone carried over from the breeding season but, rather, appeared associated with dehydroepiandrosterone, a hormone that changes rates of aggression in non-breeding birds without generating the detrimental effects of high testosterone titres that control aggression in the breeding season. This extraordinary pre-moult aggression seems puzzling because individual buntings do not hold defined territories during their moult. We speculate that this high aggression evolved as a means of regulating the number of conspecifics that moulted in what were historically small habitat patches with limited food for supporting the extremely rapid moults of painted buntings.

## 1. Introduction

In their citation classic, Fretwell & Lucas [1] use graphical models to describe how social behaviour distributes organisms across habitats and landscapes according to ideal free and ideal despotic distributions. In the ensuing 50 years, the ideal free distribution

has received by far the most attention. Nonetheless, several cases of ideal despotic habitat distributions have been documented at local, landscape and even range-wide scales. In the polygynous dickcissel, *Spiza americana*, Fretwell & Calver [2] showed, at a local scale, that male densities were higher in clover fields than pastures and that males in clover fields attracted more females than those in pastures. More surprisingly, they further showed that, across the east–west breadth of the dickcissel breeding range, male densities and levels of polygyny were highest near the centre of their breeding range in the Mississippi river valley, and that both density and mating success declined monotonically towards the eastern and western limits of the range.

More recently, age ratios, densities and settlement patterns have revealed landscape or range-wide effects of despotic territorial behaviour in black-throated blue warblers, *Setophaga caerulescens*, [3] and in hermit and Townsend's warblers, *Setophaga occidentalis* and *Setophaga townsendi*, [4,5]. In these cases, adult males occupied higher quality sites, often in higher densities, while younger first-time breeding males occupied lower quality, lower density sites consistent with despotic interactions driving settlement patterns across landscapes. While aggressive interactions should occur between adult males and between adult and second-year males, it is the displacement of second-year males across geography/habitat that can be used to infer landscape-scale effects of ideal despotic habitat distributions in the warbler examples above. Finally, a particularly compelling case of despotic behaviour driving local and long distance settlement patterns has been developed for American redstarts, *Setophaga ruticilla*, using stable isotopes to infer dispersal between natal and first breeding locations [6]. In yearling male redstarts, the timing of spring departure from their Jamaican wintering grounds was driven by habitat distribution that was despotic. Young males that wintered in the more productive mangrove habitat fattened and departed earlier in spring than those forced to winter in less productive upland habitats. After the spring departure of redstarts that had wintered in the mangroves, those that wintered in the upland habitats moved into mangroves to fatten for migration [7]. As a result, early migrating yearling males that wintered in Jamaica settled, on average, south of where they were hatched in eastern North America, while late migrating yearling males settled north of where they were hatched.

All of the above examples represent territorial social systems where the benefits of territorial defence accrue to the successful individual holder of the territory. However, many birds live in flocks during the non-breeding season and do not defend territories. Thus, aggressive behaviour in these flocking systems seems counterintuitive because benefits from an individual's aggression may also extend to other flock members. If the benefits of aggression are distributed evenly across flock members, regardless of their contribution, then non-aggressive individuals should benefit without bearing the costs of aggression [8]. It is precisely because of this potential free-loader problem that we were surprised to observe extreme levels of aggressive interactions in flocks of painted buntings, *Passerina ciris*, just after they had migrated to coastal northwest Mexico to undergo their annual moult [9,10]. For a brief period before initiating the annual moult, aggression in painted buntings is extraordinarily intense. This paper summarizes observations of aggressive encounters in pre-moulting painted buntings and offers possible explanations for the evolutionary context that would favour such behaviours.

## 2. Natural history

Like the majority of passerines breeding at lower latitudes in western North America [9,11], painted buntings from the mid-western breeding population (but not the eastern breeding population, [12]) migrate after breeding to northwest coastal Mexico for their annual moult [10]. A late summer monsoon delivers most of the annual precipitation to coastal northwest Mexico in the months of July, August and September [13]. These rains are often patchily distributed, but good rains, even if very local, result in an immediate greening of the vegetation and a huge flush of new growth and insects. Once buntings arrive in northwest Mexico, they seek areas rich with food from monsoon rains and, within these areas, buntings are abundant in weedy edges of agricultural fields and in mesquite flats that are open enough to provide grass seeds for foraging. For a brief time period between arrival on the moulting grounds to the initiation of moult, buntings become extremely aggressive, presumably to secure sufficient food resources to support their moult. During this time, we suspect that buntings move between patches to assess potential moulting sites, and that aggression may be sorting buntings across the landscape based on competitive ability and habitat quality, similar to observations in other species [3,4,7].

Moult is exceptionally intense and rapid, taking adult males about 34 days and adult females about 30 days to complete [10]. At peak moult adults probably cannot disperse to alternate moulting sites because they are growing many flight feathers, all rectrices and the majority of their body feathers

simultaneously [10]. Selecting moulting sites with sufficient food resources to support the moult is crucial because the feathers grown during these intense moults are the same feathers used for attracting mates and signalling to conspecifics during the following breeding season. By early October, no seed can be found on the Johnson grass (*Sorghum halepense*) inflorescences, a favoured food of painted buntings, and all buntings have departed northwest Sinaloa for more southern localities along the west coast of Mexico (V. G. Rohwer and S. Rohwer 2011, personal observation). Abundance indices, based on the aggregate record of collecting in Mexico, suggest that the huge numbers of painted buntings that moult in coastal northwest Mexico continue south and east, following the greening of habitats around Mexico during winter. After moulting, buntings move down the west coast of Mexico, then spill over the transvolcanic belt in winter, and migrate up the east coast of Mexico in spring [14].

Second-year male painted buntings breed in a green, female-like plumage in their first potential breeding season, as do many other North American passerines that are highly sexually dichromatic [15]. They also migrate to northwest Mexico for their first post-breeding moult in this green plumage and, thus, cannot readily be distinguished in the field from adult females. During this moult, second-year males acquire their first painted plumage, which they carry for the remainder of winter and the upcoming breeding season; similarly, older males already in their painted plumage renew these feathers and carry them until the following late summer moult.

We first discovered impressive numbers of painted buntings that seemed crazed by aggression in early August 2005, between Los Mochis and El Fuerte, Sinaloa. Buntings were foraging in field edges and using adjacent mesquite patches for resting and shade. Along the larger field edges, buntings numbered in the hundreds. Adult males were everywhere and displacing each other in numerous flight chases. But the surprise was that we never observed aggressors taking over the grass head of the aggressee. Instead, displaced birds were pursued, as though being harassed; thus, displacers seemed to be wasting energy in pursuits, rather than gaining immediate access to foraging sites. Indeed, there were always many unoccupied seed heads that could have been used for foraging, without displacements. Adult male buntings also frequently chased rough-winged swallows, *Stelgidopteryx serripennis*, blue-black grassquits, *Volatinia jacarina*, common ground doves, *Columbina passerina*, and other species as they were flying out to the weedy field edges to forage.

Nothing we could see about this aggression seemed possible to interpret as offering immediate benefits to the aggressor. Aggressive individuals bear the costs of pursuits and of elevated predation risk during these chases. If aggressive individuals succeed in displacing individuals to other areas, then they should benefit through reduced bunting densities or changes in flock composition, such that flock members have similar competitive abilities; both outcomes should help to secure food resources for moulting within a patch, or reduce the risk of density-dependent predation or other processes. However, these outcomes are contingent on interactions with buntings that do not engage in aggressive pursuits. Not only should these 'free-loaders' benefit from reduced densities, but they do not bear costs associated with aggressive pursuits [8]. Presumably, the costs to free-loaders of chases or interrupted feeding from aggressive individuals do not outweigh the benefits gained by remaining in higher density, food-rich areas. These possible costs and benefits, however, remain to be tested in this system.

To explore this puzzling aggression, we first observed bunting interactions before and after the start of moult to determine if this aggression was despotic or random between bright males and green birds. We use the term 'despotic' as a collective for an age class of birds, similar to other studies [3,4]. Specifically, we are referring to adult males as a despotic phenotype that should chase both adults and green birds, but rarely be chased by green birds; we did not assess the aggressiveness of individual birds because we did not mark buntings with unique identifiers. Next, we assessed habitat quality to see how aggressive interactions might distribute buntings across the landscape and tested for differences in bunting densities and age-class ratios between high- and low-quality sites. Finally, we explored mechanisms underlying this aggressive behaviour. In breeding birds, increases in testosterone result in aggressive territorial behaviour in males [16], similar to the behaviours we observed in buntings immediately before their moult. Because painted buntings undergo a relatively short migration from their breeding grounds to their moulting grounds in northwest Mexico, the physiological changes associated with migration (i.e. changes in hormone profiles) may not be as pronounced as those observed in long distance migrants [17]. Thus, high testosterone levels carried over from the breeding range could be responsible for the high rates of aggression we observed. However, elevated levels of testosterone are known to trade-off with immune function and also suppress moult in many migrants, thus the aggression we observed could be related to other hormones, such as dehydroepiandrosterone (DHEA), that can modulate aggression without creating the trade-offs observed with high plasma testosterone levels [18]. To better understand the potential mechanisms responsible for this aggression, we examined hormone profiles: (i) shortly

after buntings arrived in northwest Mexico, (ii) during peak aggression, and (iii) during the moult when aggression was low.

# 3. Methods

## 3.1. Measuring aggression

We measured aggression in 2005 and 2006 by picking any painted bunting that was either flying or perched where it could be seen, and recording the time it was observed and whether it displaced another bird or was displaced. Because the vegetation, which consisted mainly of Johnson grass, was tall and rank, following focal birds for long periods of time was impossible because they quickly settled into the grass where they were not visible. Most interactions were flight chases from buntings moving between shady resting sites in mesquite forest edges to foraging sites in grassy fields; flight distances were typically short (less than 50 m) and these chases lasted only seconds before focal birds perched in tall grass and disappeared from view. During observation periods, we watched buntings for 2–4 days, spanning 3.75–8.75 h. To calculate rates of aggression, we used only the time that buntings interacted with each other (i.e. the summed duration of flight chases or perch chases). Thus, despite the apparently short observation durations summarized in tables, these observations capture many bunting interactions. We scored all focal buntings as either an adult male in breeding plumage, meaning it was an 'after second year' (ASY) bird (indicating this bird has spent at least two seasons on the breeding grounds), or as a green bird. Most green birds were 'second year' (SY) males (indicating that this bird had just finished its first breeding season), though some surely were early migrant females. ASY and SY males arrive in this area at about the same time, while most, but not all, females arrive considerably later. Hatch-year (HY) birds arrive much later. Thus, birds scored as green could have been SY males, females or, later in the season, HY individuals (figure 1).

We made all observations of bunting displacements at high-quality sites (Tehueco and Tesila, see below for site descriptions), because buntings were in high densities and always interacting. We did not assess aggression at the low-quality site (El Naranjo) because bunting interactions were too infrequent.

We assessed aggression for three time periods: (i) as soon as buntings arrived to northwest Mexico, (ii) after buntings had settled but before the initiation of moult, and (iii) during the moulting period. For 2005, we do not have an aggression rate for the arrival period because we found this area in August after large numbers of painted buntings had already arrived here. For 2006, we arrived at this same locality in late July as bunting densities were increasing, so our aggression indices for this year include a period of arrival. Even though chases of other species were frequently observed in both years, our summary of aggression excludes those pursuits because our focus was on how this aggression affected the habitat distribution of painted buntings. It seems unlikely that the other species buntings frequently chased were responding by dispersing, and we made no attempt to collect data addressing this point.

## 3.2. Measuring sex and age classes across different habitats

We assessed densities and ratios of sex and age classes across habitat qualities to test for possible effects of aggression influencing settlement locations of arriving buntings. We calculated ratios of the various sex and age classes using net captures, and we measured densities as the numbers of birds captured per net-hour. Part of our work in northwest Mexico was to collect good samples of the various sex and age classes of painted buntings; thus, we estimated the relative arrival dates of ASY males, SY males, female breeders and HY birds as the collection date of the first 20 birds in these age classes that were collected between 2005 and 2013 when we worked in coastal northwest Mexico. Netted green birds that were released could easily be assigned to the three relevant sex and age classes, SY males, adult female and HY following Pyle [19].

We measured habitat quality in 2007 in high-density habitats along agricultural fields near Tehueco (26°18.2′ N, 108°41.9′ W) or in moist mesquite flats near Tesila (26°18.3′ N, 108°48.5′ W), and in a low-density Sonora desert habitat near El Naranjo (26°16.5′ N, 108°47.7′ W), which was green from recent monsoon rains. Vegetation differences between the high- and low-density habitats were assessed in ten 50 × 50 cm randomly selected quadrats in each of the two habitats. Quadrats were chosen randomly by throwing a stick and sampling 5 m away from the stick in the direction it was pointing. Then we walked 50 m in the opposite direction and repeated the throw to choose the next sampling

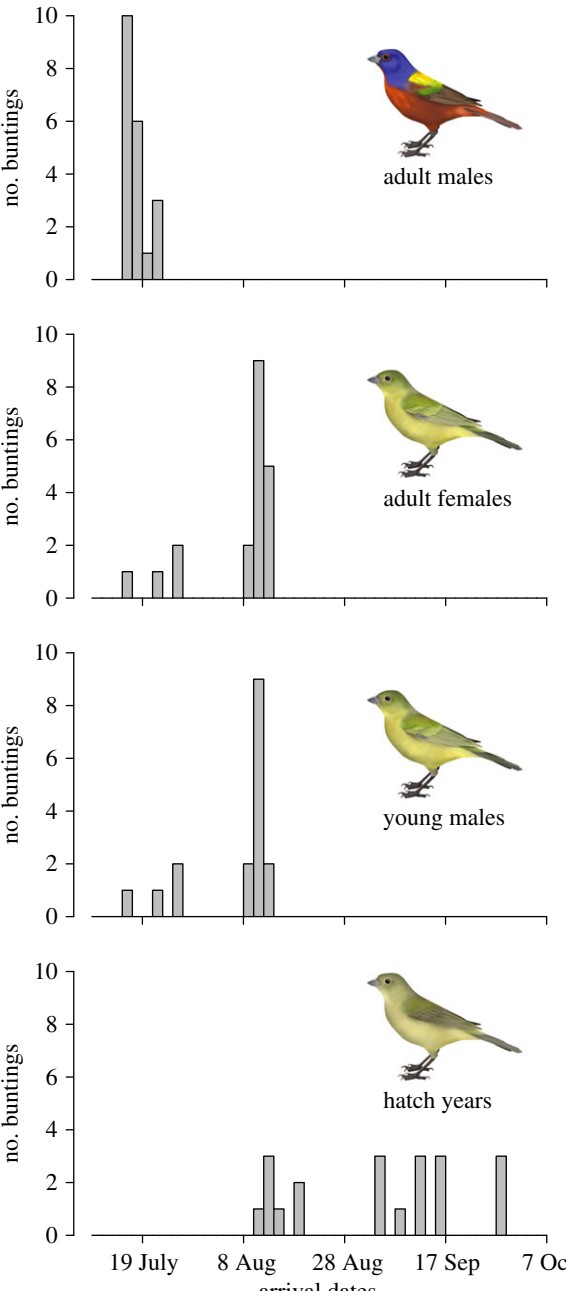

**Figure 1.** Sequence of arrival by the various sex and age classes of painted buntings in northwest coastal Sinaloa where they undergo the annual moult. Histograms are based on the first 20 individuals captures using mist-nets and show adult males arriving earliest, followed by young males and adult females, with birds hatched earlier that spring being the last to arrive. While start dates of moult vary from year to year depending on the monsoon schedule, males typically initiate moult approximately 14 days earlier than females [10].

site. In the quadrats, we assessed the above-ground weight of grasses and forbes, and the number of rocks greater than 2 cm in diameter.

## 3.3. Aggression, testosterone and dehydroepiandrosterone

We sampled blood from 75 buntings in 2006 for hormone assays to assess whether the high levels of aggression might be an artefact of high breeding testosterone levels being carried to northwest Mexico in the relatively short west-southwest migration from the mid-western breeding range of these moult-migrant buntings. We collected small blood samples by puncturing the brachial vein and collecting approximately 70–140 µl of blood into heparinized microhaematocrit tubes; all samples were taken

within 4 min of buntings hitting mist-nets. Blood samples were stored on ice until centrifuged, usually within 2 h from the time of collection. Plasma was stored in liquid nitrogen for the remainder of the field season, then transferred to −20°C freezers upon arrival at the University of Washington.

## 3.4. Hormone assays

We measured plasma levels of testosterone and DHEA using radioimmunoassays [20,21]. All plasma samples were equilibrated with approximately 2000 cpm of tritiated testosterone and DHEA to determine extraction efficiencies. Plasma was then extracted with 4 ml of freshly redistilled dichloromethane and the solvent phase was aspirated and taken to dryness under a stream of nitrogen at 40°C. Dried extracts were re-suspended in 2% ethyl acetate in iso-octane and transferred to diatomaceous earth/glycol columns. The purified testosterone and DHEA fractions were eluted in increasing concentrations of ethyl acetate in iso-octane. Eluted solvent was dried under nitrogen and extracts re-suspended in phosphate-buffered saline with gelatin. Testosterone and DHEA extracts were then measured by radioimmunoassay as described in detail by Wingfield *et al.* [20] and Soma & Wingfield [21]. All plasma levels were adjusted for per cent recovery following extraction and expressed as nanogram per millilitres.

During an initial assay, we failed to detect concentrations of testosterone or DHEA because plasma samples from individual buntings were too small. Because of this, we pooled plasma samples for two or three individuals of the same sex or age class within sampling periods. Thus, our original sample sizes were reduced to about one-third of their original value, but still represent a broad number of individuals from this population (original sample sizes: arrival period: $n = 13$, high aggression period: $n = 33$, moulting period: $n = 29$; pooled sample sizes: arrival period: $n = 5$, high aggression period: $n = 11$, moulting period: $n = 11$).

# 4. Results

## 4.1. Rates of aggression

Tables 1 and 2 summarize the extraordinary aggression we observed in 2005 and 2006, and also show that aggression drops to near zero after the initiation of moult. Thus, the high levels of chasing and displacements observed during the period between arrival and the initiation of moult by ASY males drop precipitously during the intense moult. In 2005, this drop in the rate of aggression was eight to ninefold from the pre-moult period to the period of moulting; in 2006, the drop in aggression (chases per minute) we recorded was even more extreme, from 65- to 150-fold between the pre-moulting and moulting period (tables 1 and 2).

Most aggression was initiated by ASY and SY males. In a few cases, we could see the green birds that initiated chases well enough to distinguish adult females and SY males, and all those chases were initiated by SY males. Thus, we assume that most aggression initiated by green birds involved SY males that have been through their first potential breeding season. The much later arrival of HY birds implies that essentially no displacements initiated by green birds involved HY birds (figure 1). Adult females presumably arrive later than ASY males because females lose to males in the sexual conflict over post-fledging care [22,23].

Overall, the rate of chasing was higher in 2005 than in 2006, but we suspect this is largely owing to the differences in the fields where our observations were made. In 2005, our observations were made in a more open weedy and grassy edge of a fallow agricultural field, while in 2006, our observations of aggression were made in a much higher and denser field of uncut silage sorghum. Bunting interactions were harder to see in this field, which probably accounts for the lower rate of aggression in 2006.

## 4.2. Aggression was despotic

In aggressive encounters, we recorded the plumage status of winners and losers as either ASY males or green birds. Recall that green birds were most likely SY males or adult females, and unlikely to be HY birds. We calculated expected frequencies of interactions between the colour classes of birds based on ratios of ASY males to green birds in netting samples of birds captured in the time period when our observations of aggression were being made. In 2005, during the pre-moult period of high aggression from 26 to 29 July, we captured 97 painted buntings in 111.3 net hours of effort in the same field edges where we recorded aggressive interactions. These birds fell into the following sex/age

**Table 1.** Bunting capture rates and chases in flight and from perches for 2005. (The reduction in the rate of aggression is unrelated to density changes between the pre-moult and moult periods, based on capture efforts of 111.3 net hours during pre-moult and 34.5 net hours during moult. The effort column summarizes the time spent observing bunting interactions and the number of focal birds observed for each time period. For both flight and perch chases, the minutes column summarizes the total duration of bunting interactions, both aggressive and non-aggressive, during each observation period; the chases column summarizes the number of aggressive chases only; and chases min$^{-1}$ column gives the rate of aggressive chases per time interval buntings interacted. Both flight and perch chase show similar drops in aggression from pre-moult and moulting periods.)

| | bunting captures net-hour$^{-1}$ | effort | flight chases | | | perch chases | | |
|---|---|---|---|---|---|---|---|---|
| | | | minutes | chases | chases min$^{-1}$ | minutes | chases | chases min$^{-1}$ |
| pre-moult 26–29 July | 0.845 | 4 h observation 46 focal buntings | 1.87 | 12 | 6.429 | 20.20 | 8 | 0.396 |
| moult 10–11 Aug | 0.812 | 3.75 h observation 109 focal buntings | 6.57 | 5 | 0.761 | 46.3 | 2 | 0.043 |
| reduction in aggression (pre-moult/moult) | | | 8.4-fold | | | 9.2-fold | | |

**Table 2.** Rates of chases in flight and from perches in 2006. (Aggression was high in the arrival period and increased somewhat in the pre-moult period for both flight and perch chases. Then there was a huge reduction in the rate of aggression from the pre-moult period to the moult period. Again, as in 2005, the reduction in the rate of aggression was unrelated to density changes between pre-moult and moult because captures rates were similar for the three periods. The somewhat lower capture rate for 3–5 August was related to nets being in the sun on one of those days. Column definitions are the same as those in table 1.)

| | bunting captures net-hour$^{-1}$ | effort | flight chases | | | perch chases | | |
|---|---|---|---|---|---|---|---|---|
| | | | minutes | chases | chases min$^{-1}$ | minutes | chases | chases min$^{-1}$ |
| arrival 24–26 July | 1.579 | 8.75 h observation 102 focal buntings | 7.38 | 9 | 1.22 | 18.12 | 2 | 0.11 |
| pre-moult 3–5 Aug | 0.807 | 4.2 h observation 107 focal buntings | 5.3 | 11 | 2.08 | 52.42 | 16 | 0.31 |
| early (arrival and pre-moult pooled) | 1.01 | 12.95 h observation 209 focal buntings | 12.68 | 20 | 1.58 | 70.54 | 18 | 0.26 |
| moult 19–21 Aug | 1.250 | 4.1 h observation 127 focal buntings | 3.77 | 3 | 0.014 | 218.9 | 1 | 0.0046 |
| reduction in aggression (early/moult) | | | 112.9-fold | | | 56.5-fold | | |

categories: 47 ASY males; 20 SY males; 26 adult females; and 0 HY birds. We computed expected frequencies of interactions using the 47 ASY males and the 46 green birds because only ASY males and green birds could always be distinguished in the rapid pace of recording aggressive interactions. Expected frequencies of displacements, computed from a binomial expansion, showed most

**Table 3.** Displacement patterns in 2005 during the pre-moult period of heightened aggression. (Expected frequencies were computed from netted ratios of ASY (males with blue heads, red bellies and yellow-green backs) and green birds (uniformly yellow-green plumage throughout body). Aggression was significantly despotic ($\chi^2 = 12.5$; $p < 0.001$), owing to a complete absence of green birds displacing ASY males and an excess of ASY males displacing ASY males.)

| | expected frequencies | expected number | observed number | obs — exp |
|---|---|---|---|---|
| 🐦→🐦 | 0.186 | 3.53 | 5 | 1.47 |
| 🐦→🐦 | 0.245 | 4.66 | 10 | 5.34 |
| 🐦→🐦 | 0.245 | 4.66 | 0 | −4.66 |
| 🐦→🐦 | 0.324 | 6.16 | 4 | −2.16 |

**Table 4.** Displacement patterns in 2006 during the arrival and pre-moult periods when chases were frequent. (Expected frequencies were computed from netted ratios of ASY and green birds. Again aggression was significantly despotic ($\chi^2 = 31.8$; $p < 0.001$), mainly owing to an excess of ASY males displacing each other and a complete absence of green birds displacing ASY males.)

| | expected frequencies | expected number | observed number | obs — exp |
|---|---|---|---|---|
| 🐦→🐦 | 0.255 | 9.69 | 24 | 14.31 |
| 🐦→🐦 | 0.250 | 9.5 | 7 | −2.5 |
| 🐦→🐦 | 0.250 | 9.5 | 0 | −9.5 |
| 🐦→🐦 | 0.245 | 9.31 | 7 | −2.31 |

aggression to be initiated by ASY males and directed at either other adults or at green birds in 2005 (table 3).

In 2006, we started fieldwork as buntings were still arriving, but aggression was already high, so our analyses are based on pooled data for the arrival and pre-moult periods, which are shown separately in table 2. In these early periods combined, we captured 119 buntings in 103.5 net hours. These birds fell into the following sex/age categories: 74 ASY males; 31 SY males; 14 adult females; and 0 HY birds. Again, aggression was significantly despotic ($\chi^2 = 31.8$; $p < 0.001$), but this was largely driven by an excess of ASY males displacing each other and a complete absence of green birds displacing ASY males (table 4). While much of these interactions were between ASY males, there is no reason to think that ASY males should not be displacing each other in competition for the best moulting sites, even though displaced adults cannot be used to measure despotic habitat distributions through comparisons of age ratios across habitats. Indeed, we suspect that similar measures of bunting density between the pre-moult and moult periods reported in tables 1 and 2 may be generated by adults replacing green birds that have dispersed from high-quality patches; green birds should be the most sensitive to changes in density as a result of aggression because they are the subordinate phenotype.

## 4.3. Habitat distributions of age classes reflect despotic aggression

Bunting densities, measured as captures per net-hour, were assessed in September, after all breeding male and most breeding females had initiated moult, and also after most HY birds had arrived (table 5). The capture rate of ASY males was nearly threefold higher in the high-density habitat. SY males and adult females were also captured at a 1.2-fold higher rate in the better habitat, while HY birds were captured 1.27 times more frequently in the lower density habitat, even though most HY birds arrived after aggression from adults had dropped to very low levels. HY birds may have chosen to settle in the poorer habitat simply because bunting density was lower there, or because food, while more

**Table 5.** Relative abundance of the sex/age classes netted in high- and low-density habitats from 9 to 17 September 2007, the same year habitat differences were documented. (Assessment of age/sex classes in September ensures that shifts in bunting densities as a result of pre-moulting aggression are finished, but occurs before SY males have moulted into their first painted plumage. The equal capture rates for three of the four sex/age classes in the high-density habitat is coincidence.)

| net hours | bunting density | capture rate | | | |
|---|---|---|---|---|---|
| | | adult males | subadult males | adult females | hatch-year |
| 94.0 | high | 0.340 | 0.064 | 0.340 | 0.340 |
| 136.75 | low | 0.117 | 0.051 | 0.278 | 0.658 |
| | relative abundance (high/low) | 2.91 | 1.25 | 1.23 | 0.52 |

**Table 6.** Differences in habitat quality between the high and low-density bunting habitats in 2007. (We calculated differences between sites using 10 randomly placed quadrats (50 × 50 cm) within each site, and by counting rocks and above-ground mass of grasses and forbs within each quadrat; values show means ± s.e. High-density habitat, like Tesila, had more grasses which presumably support the intense moult of buntings.)

| | no. rocks (>2 cm) | grass weight (above ground) | forb weight (above ground) |
|---|---|---|---|
| high density (Tesila) | 4.1 ± 1.7 | 54.1 g ± 9.9 | 46.4 g ± 22.3 |
| low density (El Naranjo) | 28.6 ± 10.3 | 9.1 g ± 3.4 | 32.5 g ± 10.4 |
| two-tail $p$ (median test) | 0.03 | 0.001 | 0.34 |

dispersed, was possibly less depleted in the low-density habitat. (Johnson grass, with its larger seeds and substantial seed heads was found only in the high-density habitat.)

Our quantification of differences in habitat quality were consistent with the differences in bunting densities. The high-density habitat had fewer rocks greater than 2 cm in diameter, much greater weight of grasses and somewhat more forbs (table 6).

## 4.4. Aggression and testosterone

Testosterone concentrations were so low across all three sampling periods that they could not be detected in plasma samples that are perfectly adequate for testosterone detection in temperate breeding passerines [24]. However, we measured an increase in plasma concentrations of DHEA from the arrival period to the high aggression period. DHEA concentrations remained high in the moulting period, but variation in plasma concentrations of DHEA also increased during this time (figure 2). Thus, while the increase in DHEA between arrival and pre-moulting periods corresponds to an increase in levels of aggression, DHEA did not decrease as rates of aggression did, suggesting that concentrations of DHEA alone are not responsible for differences in levels of aggression.

# 5. Discussion

Our data document several key features of this remarkable aggression in pre-moulting painted buntings. First, aggression appears despotic. ASY males chased other adults or (especially) other subordinate green birds (mostly SY males or adult females) far more than expected, based on their frequencies from contemporaneous netting. Further, green birds almost never chased ASY males. Second, this despotic aggression probably generated patterns in the density and distribution of bunting age classes between high- and low-quality sites that were consistent with an ideal despotic distribution [1]. In the better habitat, densities were higher, as assessed by birds captured per net-hour, and there were more ASY males and fewer subordinate buntings. By contrast, in the lower quality habitat, densities were lower and there were relatively more green birds. Quality of these moulting sites probably translates to

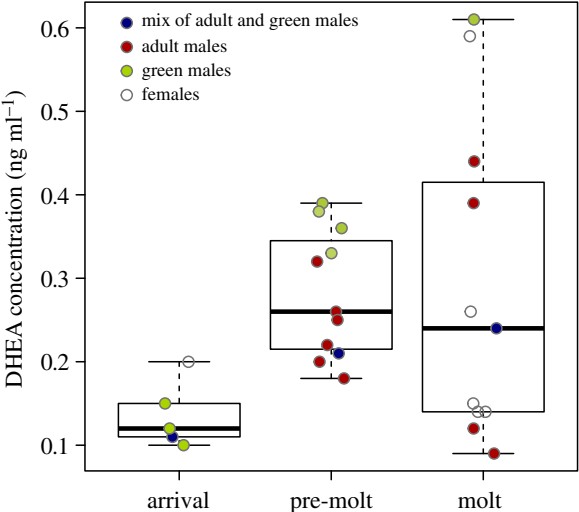

**Figure 2.** DHEA assays from blood sampled in 2006 across three time periods: arrival, pre-moult and moult. The arrival period corresponds to buntings arriving in northwest Mexico from the breeding range, pre-moult corresponds to the peak of aggression and moult corresponds with the dramatic reduction in aggressive interactions. DHEA clearly increases between arrival and pre-moult periods but does not decrease during the moulting period, suggesting that factors other than DHEA concentrations influence levels of aggression.

immediate or long-term fitness effects of having sufficient food resources to complete the moult, moulting in safe (i.e. low predation) sites, or growing bright, strong, high-quality feathers for the subsequent breeding season [25–27]. Third, the extraordinary rate of aggressive encounters dropped by 10- to more than 100-fold during the moulting period, when displacements practically ceased. The rate of aggression we observed in buntings was unlike anything we have observed in other flocking social systems, such as Harris' sparrows, *Zonotrichia querula*, dark-eyed juncos, *Junco hyemalis*, or other winter flocking sparrow species. Finally, hormone assays showed this aggression could not be attributed to high plasma testosterone levels, characteristic of summer territorial passerines, being carried to the moulting grounds (e.g. [28–30]) because testosterone levels were undetectable in our plasma samples from these moult-migrant buntings after they arrived in northwest Mexico. DHEA, which can also modulate aggression, increased from arrival to the pre-moult period of intense aggression; however, DHEA titres remained high during moult, suggesting that concentrations of DHEA alone are not responsible for the drop in aggression. Taken together, we suggest that the extreme levels of aggression observed just weeks before the onset of the annual moult help moderate bunting densities within individual food patches to ensure sufficient food resources for the upcoming moult.

The extreme level of aggression by ASY male painted buntings soon after they arrive on their moulting grounds raises intriguing evolutionary questions. In territorial social systems, the benefits of territorial defence mostly accrue to the territory holder, making the evolution of territorial behaviour unsurprising from an individual perspective. But painted buntings live in loose and fluid flocks during their moult in northwest Mexico, and benefits of this aggression are not immediately accruing to dominant individuals displacing subordinates from food resources. Thus, any benefits gained by aggressive individuals will be shared across all other birds in the interacting group, regardless of whether or not they are aggressive towards conspecifics. How then can this aggression be interpreted from an evolutionary perspective?

At least two scenarios involving multi-level selection appear consistent with our bunting observations: Wilson's [31] trait group selection model and Eldakar & Wilson's [32] selfishness as a second-order altruism model. In Wilson's [31] trait group selection model, weak altruism generates strong group selection, even with random assortment of altruists and free-loaders into groups. The key assumption of this model is that altruists benefit from their behaviour, despite its costs, even though free-loaders benefit more by not behaving aggressively. Under this scenario, we predict that a subset of ASY males should be free-loaders, as patterns of aggression indicate that adults are rarely chased or pursued by green birds, thus increasing their chances of remaining in high-quality patches without engaging in aggressive chases. By contrast, another subset of ASY males and most green birds should be altruists and contribute to the defence of food resources within patches. Weak altruism is defined as $r > d > 0$, where $r$ is the benefit of an altruist's behaviour on others and $d$ is the

benefit of the altruist's behaviour on itself. Because the relative fitness of free-loaders (in this case, non-aggressive probably ASY male buntings) is higher than that of altruists (in this case, aggressive buntings, probably both ASY males and green birds), the fixation of aggression that yields no direct benefit occurs when trait groups containing a higher frequency of altruists contribute more to the pooled population than those containing a lower frequency of altruists [33]. Wilson's [31] formulation of weak altruism falls into the case of class II fitness structures, defined and graphically represented in Kerr & Godfrey-Smith [33] and Kerr *et al.* [34] as $r > d$; cases of class II altruism result in fixation of altruistic behaviour. In short, it is the increased productivity of trait groups with more altruists, and the fact that trait groups dissolve into a larger panmictic population, that enables the fitness structure $r > d$ to fix the altruistic genotype, despite the lower relative fitness of aggressors within trait groups [31,33,34].

In Eldakar & Wilson's [32] selfishness as a second-order altruism model, individual groups are defined and maintained by a balance of group size and the frequency of 'selfish punishers' that moderate the frequency of purely selfish individuals (i.e. free-loaders). Under this model, as the frequency of free-loaders within a group rises, the fitness of both selfish punishers and altruistic individuals declines. Thus, both selfish punishers and altruistic individuals have an incentive to punish free-loaders, but individuals that were originally selfish (i.e. the 'selfish punisher'), and thus have a fitness advantage over altruists, should be in a better position to punish free-loaders. We would again predict that ASY males would be free-loaders, but ASY males should also be the most common 'selfish punisher' because they are the competitively dominant phenotype. Another subset of ASY males and most green birds should be altruists in defending food-rich patches. Importantly, for both models, we consider altruistic behaviour only with respect to the defence of food resources within a patch. Individuals that leave a patch could be considered altruists, but we suspect that individuals make these decisions from a costs–benefit perspective, such that once the costs of continued harassment, chases and interruptions during foraging exceed the benefits of remaining in a patch, these individuals disperse. These costs apparently accrue most strongly to green birds, as this phenotype is competitively subordinate to ASY males and occurs at lower frequencies in sites dominated by ASY males. Similar to Wilson's [31] trait group model, the key assumption in Eldakar & Wilson's [32] selfishness as a second-order altruism model is that any altruistic behaviour brings a within-group fitness benefit at a relatively small cost to the individual, creating fitness differentials between groups. However, once the frequency of free-loaders within a group exceeds a certain size, the benefits awarded to the group do not outweigh the costs incurred to the individual when punishing free-loaders and fitness differentials between groups disappear, and group stability dissolves [32]. Thus, the stability of groups depends on both group size and the frequency of both selfish punishers and free-loaders, because selfish punishers reduce the frequency of free-loaders and enable altruists to persist, promoting group stability. Until fitness measures of individuals and groups of moulting buntings are made, we cannot distinguish between traditional individual- and multi-level selection frameworks for the evolution of this extraordinary aggression we observed in these pre-moulting buntings.

At least two additional factors—behavioural syndromes [35] and non-adaptive by-products associated with migration or moulting—could underlie some of the variation in aggression we observed in buntings. If aggression is a behavioural syndrome, it seems strongly tied to phenotype because ASY males are by far the most aggressive. While behavioural syndromes are not an evolutionary explanation for this aggression, they may help explain why some individuals within a sex or age class are more aggressive than others, and therefore influence an individual's decision to be an aggressive defender of food patches, a free-loader or a selfish punisher. To explore this alternative, individual buntings would have to be marked and observed multiple times to determine if there is behavioural consistency in similar social contexts, and if this aggressive behaviour is associated with other behaviours, thus qualifying as a syndrome [35]. The second factor—aggression is a non-adaptive by-product of migration or moulting—seems unlikely because we know of no other examples of such extreme aggression immediately before the moult in other species. Once moult began, aggression dropped, suggesting that aggression is tied only to the pre-moulting period, during which time birds are unencumbered by feather loss and growth, and can affect densities. Because feathers cannot be repaired, selection to decouple moult from by-product activities, such as aggression, which could harm growing feathers, should be strong. Further, the pre-moult aggression in buntings is not a result of testosterone carried over from the breeding season but appears associated, in part, by increases in DHEA. In short, we doubt this aggression represents a non-adaptive by-product, but we cannot yet rule out this alternative.

The habitat distribution and the timing and ecology of moult in painted buntings in northwest Mexico seem consistent with multi-level selection models in several ways. First, the unpredictability of the late

summer monsoon generates patches of productive habitat that are variable in size, often within landscapes that have yet to receive rain and still are leafless from the preceding six to nine months of drought [13]. Second, even within areas that have received rain, grassy swales that generated seeds are patchy, further dividing the larger rainfall areas into smaller groups. Third, the moult itself is brief and intense and individuals vary so much in start dates that some may be completing their moult before others arrive and initiate moult [10]. Fourth, ASY males are the most aggressive phenotype, and they arrive earliest and presumably settle in the best habitats. Thus, once selection began favouring the aggression we observed, that aggression would reinforce the segregation of aggressors and free-loaders, creating larger than binomial variance in the frequency of aggressors and free-loaders between trait groups. Fifth, the early settlement of ASY males in the best patches makes it seem plausible that later arriving buntings (and especially those of subordinate sex and age classes) might settle in less productive and lower density habitats with sufficient resources to support their moult. Such food/density assessments, if they occur, could greatly reduce the detrimental effect of low aggression in moulting adults that cease defending habitat patches once they initiate moult. Finally sixth, the extreme intensity of the moult, which renders adults (but not HY birds) nearly flightless, must largely or completely prevent dispersal from patches where adults settled to moult, thus reinforcing the integrity of groups for the duration of the adult moult.

Why might such extraordinary aggression prior to moulting have evolved in painted buntings? Buntings have very rapid and intense moults that render them nearly flightless. They also show incredible variation in the timing of moult, with early breeding males arriving on the moulting grounds first, followed by later breeding males, then females; last to arrive on the moulting grounds are HY birds that do not undergo a complete moult [10]. This intense moult is fuelled primarily by ephemeral grass seeds. If too many individuals occupy one patch and deplete food resources before the moult is complete, the intensity of the moult would prevent buntings from dispersing to other food-rich patches. Thus, high aggression prior to moulting may have evolved to lower densities of individuals occupying a patch. Similarly, the intense moults of buntings may have evolved to help them complete the moult before later arriving birds can increase in numbers sufficiently to risk exhausting food resources before the moult can be completed. Observations by S.R. and V.G.R. in October 2011 were consistent with this claim. In several days of field observations in the same areas and habitats where, in previous years, buntings had been abundant, we failed to see even a single painted bunting. Yet, we found them in reasonable numbers further south near Mazatlán, including some green birds that were still moulting. Moreover, during these October observations, no seed whatsoever could be found in the Johnson grass heads along field edges, suggesting that late arrivals had been forced to overfly these food-exhausted moulting grounds to moult further south.

Historically, bunting moulting sites in northwest Mexico were probably confined to small weedy patches and edges that naturally limited the number of individuals occupying a patch. Aggressive behaviours probably benefited individual aggressors (as well as others buntings within a patch) by lowering bird densities within that patch. At broader spatial scales, aggression probably distributed individuals across habitats into groups based on similar aggression levels. Because of the small patch sizes, free-loaders in high-quality habitats occupied by aggressive adults were probably rare. However, the advent of massive irrigation and industrial agriculture in this region, following the land reforms of the 1960s [36], has generated field edges suitable for moulting that are surely much larger than historic habitat patches. Because these field edges can accommodate large numbers of buntings, free-loaders might more easily persist in these larger groups, reducing the presumed historical effectiveness of trait group selection models. Thus, large-scale irrigated agriculture now may be creating such large groups of buntings that the benefits of high aggression prior to moulting no longer exceed its costs.

The extreme aggressive behaviour we have described here deserves further study from a variety of perspectives. New insights into this system might come from observations using enclosures that manipulate group size and composition (ratio of ASY males to green birds) to explore whether free-loaders exist among the early arriving ASY males, and whether aggressiveness is plastic in response to group composition. Neither is predicted under the class II fitness structure of Wilson's [31] trait group selection model that we think applies to our observations [33]. By contrast, if this aggression is consistent with Eldakar & Wilson's [32] selfishness as a second-order altruism model, we would predict that aggression should be dependent on group composition and increase as the frequency of purely selfish individuals increases. From a proximate perspective, understanding the mechanisms of the extreme changes in aggression before and after the moult deserves study. Our data leave no doubt that testosterone, carried over from the breeding season, does not drive the extreme aggression we

observed. Instead DHEA concentrations, which increased from the arrival period to the pre-moult period, may generate expression of aggression. If DHEA stimulates aggression during the pre-moulting period, then its failure to do so during moult could be because there is an increase in the enzymes that convert DHEA to testosterone and oestradiol in specific brain regions associated with aggression (see [28]). Receptor sites for testosterone and oestradiol-17b derived from DHEA may also be modulated down once the moult is underway [29,30,37].

# 6. Conclusion

The extreme aggression in pre-moulting painted buntings resulted in the spatial separation of age classes and phenotypes, such that ASY males, the most aggressive sex/age class occupied higher quality sites compared to the other, less aggressive sex/age classes. Despotic behaviours probably translate into fitness benefits linked to feather quality and mate attraction, both of which are probably strongest for ASY males that rely on their painted plumage to attract mates and signal to conspecifics. Presumably, males that occupy high-quality moulting sites and successfully limit the number of other buntings using the same patch-specific food resources complete their intense moult without having to disperse to other locations during the moult. Yet, the aggression in group settings raises intriguing questions about levels of selection. That many features of this system appear consistent with multi-level selection models suggests more examination of flocking social systems would be instructive. In particular, species with flocks that seem fluid and that clearly are not organized into stable dominance hierarchies may have similar fitness structures that can generate strong between-group selection [8]. Regardless of the selective framework that best depicts these data, such aggressive behaviour after the breeding season and immediately before moulting, seems to be no accident and awaits further study.

Ethics. All fieldwork associated with the study was conducted under permits issued to the Universidad Nacional Autonoma de Mexico (UNAM) in collaboration with Adolfo Navarro-Siguenza, Curator of Birds at the Museo de Zoologia, Facultad de Ciencias, UNAM, and was approved by the University of Washington's Animal Care and Use Committee.
Data accessibility. All data are included in the electronic supplementary material.
Authors' contributions. S.R. designed the study, V.G.R., S.R. and J.C.W. collected and analysed the data, S.R., V.G.R. and J.C.W. wrote the paper. All authors gave final approval for publication.
Competing interests. The authors declare no competing interests.
Funding. Fieldwork in west Mexico was supported by The Burke Museum Endowment for Ornithology, a grant from the Nuttall Ornithological Club, a contract from Region 6 of the US Fish and Wildlife Service and gifts from Hugh S. Ferguson to the Burke Museum. J.C.W. acknowledges support from the Russell F. Stark University Professorship.
Acknowledgements. Chris Wood, Rob Faucett, Jessie Barry, Jeff Kelly, Samuel Lopez, Marco Ortiz and Jamie Navarro helped with fieldwork in Sinaloa. Special thanks to Ben Kerr for many helpful discussions about levels of selection, and to Megan Bishop for the wonderful bunting illustrations. Two anonymous reviewers provided comments that helped clarify sections of the paper. Specimens referred to in this paper are housed at the University of Washington Burke Museum.

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
