## [Reviewer comments · Royal Society Open Science]

Review History

RSOS-191510.R0 (Original submission)

Review form: Reviewer 1

Is the manuscript scientifically sound in its present form?

Yes

Are the interpretations and conclusions justified by the results?

Yes

Is the language acceptable?

Yes

Do you have any ethical concerns with this paper?

No

Have you any concerns about statistical analyses in this paper?

No

Recommendation?

Accept with minor revision (please list in comments)

Comments to the Author(s)

The study by Rohwer et al. set out to test possible evolutionary and physiological explanations for aggression in pre-molting Painted Buntings (*Passerina ciris*). To do this, the authors compared rates of aggression just before molt began versus during molt and estimated the distribution of age classes across different quality habitats. Adult males chased other adults and subordinates far more than would be expected based on the frequencies of each class and sex, which the authors suggest influences the distribution of age classes across different quality habitats. Furthermore, aggression was not related to testosterone because levels of this hormone were undetectable. By contrast, levels of DHEA appear to be related to levels of aggression because the increase in aggression from arrival to pre-molt was mirrored in an increase in DHEA levels. However, DHEA did not decrease during molt, when aggression did, so the authors suggest DHEA alone is not the only factor controlling aggression. Overall, the experiment is well designed, the methods are appropriate, and the manuscript is well written.

My main criticism is in regards to the introduction and some aspects of the study that are either insufficiently described or not described at all. Most importantly, in my view, the authors do not describe in the introduction what they consider to be despotic aggression. It's not until the results (line 260-264) that it becomes clear what the authors mean by this term. Second, the introduction mentions testosterone only once, and that is simply to say that it was measured. There is nothing in the introduction to explain why the authors would expect this hormone to be related to aggression. An even more extreme example is that of DHEA, which is not mentioned at all in the introduction. A short section of the introduction describing why these hormones could be expected to be related to aggression in birds seems appropriate.

Finally, could the authors provide an indication of how long elapsed between a bird being caught in a mist net and the blood sample being collected? This information is potentially important given that levels of plasma testosterone are clearly low in painted buntings during the study and that levels of this hormone have been shown to rapidly decrease due to the acute stress of capture.

Review form: Reviewer 2

Is the manuscript scientifically sound in its present form?

Yes

Are the interpretations and conclusions justified by the results?

Yes

Is the language acceptable?

Yes

Do you have any ethical concerns with this paper?

No

Have you any concerns about statistical analyses in this paper?

Yes

Recommendation?

Major revision is needed (please make suggestions in comments)

Comments to the Author(s)

File is attached (Appendix A).

Decision letter (RSOS-191510.R0)

04-Nov-2019

Dear Dr Rohwer,

The editors assigned to your paper ("Despotic Aggression in Pre-molting Painted Buntings") have now received comments from reviewers. We would like you to revise your paper in accordance with the referee and Associate Editor suggestions which can be found below (not including confidential reports to the Editor). Please note this decision does not guarantee eventual acceptance.

Please submit a copy of your revised paper before 27-Nov-2019. Please note that the revision deadline will expire at 00.00am on this date. If we do not hear from you within this time then it will be assumed that the paper has been withdrawn. In exceptional circumstances, extensions may be possible if agreed with the Editorial Office in advance. We do not allow multiple rounds of revision so we urge you to make every effort to fully address all of the comments at this stage. If deemed necessary by the Editors, your manuscript will be sent back to one or more of the original reviewers for assessment. If the original reviewers are not available, we may invite new reviewers.

- Data accessibility

<http://datadryad.org/submit?journalID=RSOS&manu=RSOS-191510>

- **Competing interests**

- **Authors' contributions**

- **Acknowledgements**

- **Funding statement**

Kind regards,
Andrew Dunn
Senior Publishing Editor
Royal Society Open Science
openscience@royalsociety.org

on behalf of Dr Claudia Wascher (Associate Editor) and Kevin Padian (Subject Editor)
openscience@royalsociety.org

Associate Editor's comments (Dr Claudia Wascher):

Associate Editor: 1

Comments to the Author:

The presented study investigates aggression in a passerine bird during pre-molting season, when birds are not territorial. Overall, the reviewers find the presented study interesting, well designed and the conclusions justified by the data, however they do provide some suggestions regarding the overall framework of the manuscript, which should be addressed prior to publication.

Comments to Author:

Reviewers' Comments to Author:

Reviewer: 1

Comments to the Author(s)

The study by Rohwer et al. set out to test possible evolutionary and physiological explanations for aggression in pre-molting Painted Buntings (*Passerina ciris*). To do this, the authors compared rates of aggression just before molt began versus during molt and estimated the distribution of age classes across different quality habitats. Adult males chased other adults and subordinates far more than would be expected based on the frequencies of each class and sex, which the authors suggest influences the distribution of age classes across different quality habitats. Furthermore, aggression was not related to testosterone because levels of this hormone were undetectable. By contrast, levels of DHEA appear to be related to levels of aggression because the increase in aggression from arrival to pre-molt was mirrored in an increase in DHEA levels. However, DHEA did not decrease during molt, when aggression did, so the authors suggest DHEA alone is not the only factor controlling aggression. Overall, the experiment is well designed, the methods are appropriate, and the manuscript is well written.

My main criticism is in regards to the introduction and some aspects of the study that are either insufficiently described or not described at all. Most importantly, in my view, the authors do not describe in the introduction what they consider to be despotic aggression. It's not until the results (line 260-264) that it becomes clear what the authors mean by this term. Second, the introduction mentions testosterone only once, and that is simply to say that it was measured. There is nothing in the introduction to explain why the authors would expect this hormone to be related to aggression. An even more extreme example is that of DHEA, which is not mentioned at all in the introduction. A short section of the introduction describing why these hormones could be expected to be related to aggression in birds seems appropriate.

Finally, could the authors provide an indication of how long elapsed between a bird being caught in a mist net and the blood sample being collected? This information is potentially important given that levels of plasma testosterone are clearly low in painted buntings during the study and that levels of this hormone have been shown to rapidly decrease due to the acute stress of capture.

Reviewer: 2

Comments to the Author(s)

File is attached.

Author's Response to Decision Letter for (RSOS-191510.R0)

See Appendix B.

RSOS-191510.R1 (Revision)

Review form: Reviewer 1

Is the manuscript scientifically sound in its present form?

Yes

Are the interpretations and conclusions justified by the results?

Yes

Is the language acceptable?

Yes

Do you have any ethical concerns with this paper?

No

Have you any concerns about statistical analyses in this paper?

No

Recommendation?

Accept as is

Comments to the Author(s)

After reading the revised manuscript (and the authors' response to the other reviewer) it is clear that the authors have carefully considered and addressed our comments. The manuscript is improved by the revisions and I have no further comments.

Review form: Reviewer 2

Is the manuscript scientifically sound in its present form?

Yes

Are the interpretations and conclusions justified by the results?

Yes

Is the language acceptable?

Yes

Do you have any ethical concerns with this paper?

No

Have you any concerns about statistical analyses in this paper?

No

Recommendation?

Accept with minor revision (please list in comments)

Comments to the Author(s)

The revised ms. RSOS-191510.R1 is much improved. The authors did overall a good job of answering reviewers' comments. They have clarified the hypothesized relationships between observations and theory, improved legends, and made all parts flow more easily. As a reader, I appreciate the effort they have made on all points.

The one substantive point that, I suggest, merits a change is in how behavioral syndromes are cast. They are not evolutionary explanations in themselves and cannot be an alternative to explanations like a multi-level selection model – or even non-adaptive explanations. But behavioral syndromes are important here, because one expects that hormone-mediated differences in aggression might be connected selectively with additional phenotypic differences e.g. in dealing with stress, or energy allocation to courtship vs aggression on the breeding grounds – and contribute to the behavioral decisions on whether to be aggressive or to freeloader in molting habitats. Thus complex behavioral variation must somehow underlie any fixed

differences between birds and their roles in molting time interactions. Multi-level selection could favor and select for distinct types or flexible types (age and condition related?); for distinct types, selection for altruistic aggression could also build on breeding-ground-selected differences in DHEA receptors, for instance, and result in behavioral syndromes.

One would then look to the few Greens chasing Greens for future altruists/aggressors, but as the authors note, this waits for individually marked birds and probably controlled conditions.

(Also note that “selfish punishment” fits within or extends multi-level selection by adding other behavioral options; it isn’t really an alternative theory. But it does create a different expectation of the link between individual and behavior – now there are potentially three types of birds.)

Decision letter (RSOS-191510.R1)

19-Dec-2019

Dear Dr Rohwer,

On behalf of the Editors, I am pleased to inform you that your Manuscript RSOS-191510.R1 entitled "Despotic Aggression in Pre-molting Painted Buntings" has been accepted for publication in Royal Society Open Science subject to minor revision in accordance with the referee suggestions. Please find the referees' comments at the end of this email.

The reviewers and Subject Editor have recommended publication, but also suggest some minor revisions to your manuscript. Therefore, I invite you to respond to the comments and revise your manuscript.

- Ethics statement

- Data accessibility

If you wish to submit your supporting data or code to Dryad (<http://datadryad.org/>), or modify your current submission to dryad, please use the following link:
<http://datadryad.org/submit?journalID=RSOS&manu=RSOS-191510.R1>

- Competing interests

- Authors' contributions

- Acknowledgements

- Funding statement

Because the schedule for publication is very tight, it is a condition of publication that you submit the revised version of your manuscript before 28-Dec-2019. Please note that the revision deadline will expire at 00.00am on this date. If you do not think you will be able to meet this date please let me know immediately.

- 1) A text file of the manuscript (tex, txt, rtf, docx or doc), references, tables (including captions) and figure captions. Do not upload a PDF as your "Main Document".
- 2) A separate electronic file of each figure (EPS or print-quality PDF preferred (either format should be produced directly from original creation package), or original software format)
- 3) Included a 100 word media summary of your paper when requested at submission. Please ensure you have entered correct contact details (email, institution and telephone) in your user account

4) Included the raw data to support the claims made in your paper. You can either include your data as electronic supplementary material or upload to a repository and include the relevant doi within your manuscript

5) All supplementary materials accompanying an accepted article will be treated as in their final form. Note that the Royal Society will neither edit nor typeset supplementary material and it will be hosted as provided. Please ensure that the supplementary material includes the paper details where possible (authors, article title, journal name).

on behalf of Dr Claudia Wascher (Associate Editor) and Kevin Padian (Subject Editor)
openscience@royalsociety.org

Reviewer comments to Author:

Reviewer: 1
Comments to the Author(s)

After reading the revised manuscript (and the authors' response to the other reviewer) it is clear that the authors have carefully considered and addressed our comments. The manuscript is improved by the revisions and I have no further comments.

Reviewer: 2
Comments to the Author(s)

The revised ms. RSOS-191510.R1 is much improved. The authors did overall a good job of answering reviewers' comments. They have clarified the hypothesized relationships between observations and theory, improved legends, and made all parts flow more easily. As a reader, I appreciate the effort they have made on all points.

The one substantive point that, I suggest, merits a change is in how behavioral syndromes are cast. They are not evolutionary explanations in themselves and cannot be an alternative to explanations like a multi-level selection model—or even non-adaptive explanations. But behavioral syndromes are important here, because one expects that hormone-mediated differences in aggression might be connected selectively with additional phenotypic differences e.g. in dealing with stress, or energy allocation to courtship vs aggression on the breeding grounds—and contribute to the behavioral decisions on whether to be aggressive or to freeload in molting habitats. Thus complex behavioral variation must somehow underlie any fixed differences between birds and their roles in molting time interactions. Multi-level selection could favor and select for distinct types or flexible types (age and condition related?); for distinct types,

selection for altruistic aggression could also build on breeding-ground-selected differences in DHEA receptors, for instance, and result in behavioral syndromes.

One would then look to the few Greens chasing Greens for future altruists/aggressors, but as the authors note, this waits for individually marked birds and probably controlled conditions.

(Also note that “selfish punishment” fits within or extends multi-level selection by adding other behavioral options; it isn’t really an alternative theory. But it does create a different expectation of the link between individual and behavior – now there are potentially three types of birds.)

Author's Response to Decision Letter for (RSOS-191510.R1)

See Appendix C.

Decision letter (RSOS-191510.R2)

07-Jan-2020

Dear Dr Rohwer,

It is a pleasure to accept your manuscript entitled "Despotic Aggression in Pre-molting Painted Buntings" in its current form for publication in Royal Society Open Science. The comments of the reviewer(s) who reviewed your manuscript are included at the foot of this letter.

on behalf of Dr Claudia Wascher (Associate Editor) and Kevin Padian (Subject Editor)
openscience@royalsociety.org

Appendix A

Review of MS RSOS 191510 Rowher et al

This study reports and interprets patterns of aggression among migrating Painted Buntings, off territory and during a period when migrants pause, feeding in flocks, while going into and through molt. They identify a potential problem in understanding the high levels of aggression just before the molt, because aggressors are displacing others, not over a particular limited food item that they can then eat themselves, but from the foraging patches, which vary in quality and by year. Thus, they argue, the benefits of lowered competition are going not just to the aggressors (who bear the energetic and time costs of aggression) but to other remaining less aggressive birds that may enjoy more food when it becomes critical during the period of molt and relative immobility. The authors point out that this potential freeloading by non-aggressors makes individual-selection based explanations of aggression problematic.

The study presents new information on the temporal pattern of aggression with respect to pre-molt and molt periods in stop-over habitats, measures of quality of the habitat patches, the changes in the distribution, sex ratios and overall abundance of buntings by the molt period, and an analysis of hormone levels in birds across these periods that might provide insight into the proximate drivers of aggression. They then highlight ways in which individual selection (individual payoffs to aggressors) is an unlikely evolutionary explanation of these patterns, especially of the aggressive phenotype of adult males during this period. They propose that multilevel selection among patches, with relative productivity of the patch depending on the degree to which abundance of remaining birds are appropriate to the amount of food to support successful molt, provides an alternative framework.

Overall, the recognition that this basic natural history observation—radically heightened aggression—poses an interesting evolutionary problem is refreshing. There is much still to be observed and investigated further to see how theories can actually explain what we see. But we do need to notice where the assumptions of a given construct are met or not. That sets this paper apart.

What the paper does less well is clearly present the questions answered with data and their a priori relation to costs and benefits as felt by the birds. The last paragraphs of the Intro stress the behavioral measures, but as an entrée to the data, we need some indication of what the authors think are the benefits and costs to both aggressors and non-aggressors (that aren't driven out). Clearly costs are being forced upon some of the freeloaders—energy of fleeing, disrupted feeding—as well as aggressors. As a reader, I would like to see a short clear possible scenario of the specific costs and benefits and their altruistic distribution, then a clear set up of the questions answered here and how they relate: Adjustments of flock size to support molt (which predicts what about repeated density assays?), changes or not to age-sex ratios if freeloaders are actually freeloading?, etc. Similarly, to follow the arguments, a time-line of bunting arrival, aggression, moves between patches, etc. would be very useful in the Natural History section, where one could make clear what aspects are known, what are predicted under various cost-benefit scenarios, and what the authors assessed in this paper.

Discussion—the basic argument that there is a freeloading class of less-non aggressive individuals is clear. What is less clear is how one would discriminate the several ways in which birds could be acting as altruists, freeloaders, selfish or unselfish individuals, punishers, etc. I try to make this problem clear below and then note some specific problems with figures and tables.

Starting **P 14**. I think that the discussion of free-loading would be a lot clearer if the authors could put it directly in the current context—could operationalize the elements of theory with actual events. Who, behaviorally, are the freeloaders, the selfish punishers and how would this scenario translate to the bunting pre-molt context? Who are the unselfish altruists in this scenario—those who leave without being pushed out and without contesting access to the best foods? This is one of several places where the authors refrain from fully translating their situation into the two theoretical constructs within a multilevel selection framework. It would greatly strengthen their argument if these translations were made clear—and the two scenarios do differ in how each would expect the bunting flock members to do cost-benefit-wise.

P 16 | 377-379 introducing alternative non-adaptive or spandrel-style explanation: The authors should not call “behavioral syndromes” a necessarily non-adaptive explanation. A behavioral syndrome is simply a description of trait structure, that can, yes, potentially limit adaptive flexibility or response to selection IF selection bumps up against features of the syndrome that are not easily unlinked. But they don’t have to limit adaptation in a given situation. I didn’t find the discussion of this particularly appropriate to behaviors that may or may not be highly variable within an age-sex class. That would await further study of individuals and the question of whether particular individual phenotypes bear the brunt of costs, but (for instance) cannot disappear because those phenotypes were also the most successful breeders.

P 16, L392—here in discussion the authors get less careful about attributing the regulation of aggression to DHEA, albeit saying “in part”. Perhaps regulating is too loaded a term.

Tables 1, 2. I don’t follow what some of the variables are. What I “Minutes” measuring? Total time spent assessing chases? If so, the time periods are sometimes VERY short (1.87, 6.57) and vary all over the place. Minutes is used in calculating rate of chases. But if minutes is total time spent counting chases, that doesn’t seem adequate as basis for an estimate. Time of day? Number of days of assessment? For strong behavioral sampling, I would have tried to equalize sample effort.

P 25 Table 5 shows age-sex classes in September, but the year is 2007; I presume this is a typo because in the text 2006 was the year in which hormones were assayed.

Also it is interesting that the actual densities had not changed by the time molt arrived, in either year. (A drop between arrival densities and pre-molt densities in 2006 is explained by net placement.) Would one not expect a drop if chasing were effective in regulating incoming bird numbers?

Table 4. There is something wrong here as the expected numbers (based on expected fractions of observations * total observations I assume) do not add up to the total observations (n=38), thus making the generated Obs-Exp incorrect. They do add up to same in Table 3 which uses the same approach.

Table 6. Need to put more info in legend—e.g. 50 x 50 cm plots, etc. Legends in general need to stand alone. Assuming that “mean differences” are what we are supposed to see from the means presented, shouldn’t there be SD or SE for these means?

Fig 1. Needs a lot more info in legend to tell reader what these data are based on. It isn’t clear when the start of molt is, after arrival. (Is there a problem with age-ratio assessment in mid-late September if some of SY males have molted into AHY plumage, adding to AHY birds that have finished?)

Fig 2. Why are the DHEA levels not separated by SY vs AHY birds? This shows an increase from arrival to pre-molt, while aggression levels at arrival in 2005 appear to be very high. And the overall DHEA levels should be very different between SY and AHY birds, if related closely to aggressive chases.

Appendix B

25 November 2019

Dear Editors of Royal Society Open Science,

We are re-submitting a revised version of our manuscript RSOS-191510, Despotic aggression in pre-molting Painted Buntings. Both Reviewers provided constructive comments to improve clarity and readability of the paper. We have incorporated these comments in this revised version and provide a summary of these changes below; Reviewer comments are in regular font and our response to comments are in *italics*.

We think this revised version is much improved and hope you and Reviewers find this improved draft suitable for RSOS readers.

Best wishes,

Vanya Rohwer, Sievert Rohwer, John Wingfield

Response to Referees

Reviewer 1

The study by Rohwer et al. set out to test possible evolutionary and physiological explanations for aggression in pre-molting Painted Buntings (*Passerina ciris*). To do this, the authors compared rates of aggression just before molt began versus during molt and estimated the distribution of age classes across different quality habitats. Adult males chased other adults and subordinates far more than would be expected based on the frequencies of each class and sex, which the authors suggest influences the distribution of age classes across different quality habitats. Furthermore, aggression was not related to testosterone because levels of this hormone were undetectable. By contrast, levels of DHEA appear to be related to levels of aggression because the increase in aggression from arrival to pre-molt was mirrored in an increase in DHEA levels. However, DHEA did not decrease during molt, when aggression did, so the authors suggest DHEA alone is not the only factor controlling aggression. Overall, the experiment is well designed, the methods are appropriate, and the manuscript is well written.

My main criticism is in regards to the introduction and some aspects of the study that are either insufficiently described or not described at all. Most importantly, in my view, the authors do not describe in the introduction what they consider to be despotic aggression. It's not until the results (line 260-264) that it becomes clear what the authors mean by this term. Second, the introduction mentions testosterone only once, and that is simply to say that it was measured. There is nothing in the introduction to explain why the authors would expect this hormone to be related to aggression. An even more extreme example is that of DHEA, which is not mentioned

at all in the introduction. A short section of the introduction describing why these hormones could be expected to be related to aggression in birds seems appropriate. *We agree with both comments. In the introduction, we now defined what we consider despotic aggression (in this case, we're referring to an age and sex -- adult males -- as a despotic phenotype), and outlined why we examined hormone profiles of buntings to better understand aggression. We briefly describe the role of testosterone for regulating aggression during the breeding season and why high T levels might be carried over to the molting grounds in this short distance migrant. We also briefly outline trade-offs associated with high T levels and introduce other hormones, like DHEA, as an alternative to modulating aggression without the costs of high T levels.*

Finally, could the authors provide an indication of how long elapsed between a bird being caught in a mist net and the blood sample being collected? This information is potentially important given that levels of plasma testosterone are clearly low in painted buntings during the study and that levels of this hormone have been shown to rapidly decrease due to the acute stress of capture.

We state in the methods that all birds were sampled in under 4 minutes from the time of capture in mist nets.

Reviewer 2

This study reports and interprets patterns of aggression among migrating Painted Buntings, off territory and during a period when migrants pause, feeding in flocks, while going into and through molt. They identify a potential problem in understanding the high levels of aggression just before the molt, because aggressors are displacing others, not over a particular limited food item that they can then eat themselves, but from the foraging patches, which vary in quality and by year. Thus, they argue, the benefits of lowered competition are going not just to the aggressors (who bear the energetic and time costs of aggression) but to other remaining less aggressive birds that may enjoy more food when it becomes critical during the period of molt and relative immobility. The authors point out that this potential freeloading by non- aggressors makes individual-selection based explanations of aggression problematic.

The study presents new information on the temporal pattern of aggression with respect to pre- molt and molt periods in stop-over habitats, measures of quality of the habitat patches, the changes in the distribution, sex ratios and overall abundance of buntings by the molt period, and an analysis of hormone levels in birds across these periods that might provide insight into the proximate drivers of aggression. They then highlight ways in which individual selection (individual payoffs to aggressors) is an unlikely evolutionary explanation of these patterns, especially of the aggressive phenotype of adult males during this period. They propose that multilevel selection among patches, with relative productivity of the patch depending on the degree to which abundance of remaining birds are appropriate to the amount of food to support successful molt, provides an alternative framework.

Overall, the recognition that this basic natural history observation—radically heightened aggression—poses an interesting evolutionary problem is refreshing. There is much still to be observed and investigated further to see how theories can actually explain what we see. But we do need to notice where the assumptions of a given construct are met or not. That sets this paper apart.

What the paper does less well is clearly present the questions answered with data and their a priori relation to costs and benefits as felt by the birds. The last paragraphs of the Intro stress the behavioral measures, but as an entrée to the data, we need some indication of what the authors think are the benefits and costs to both aggressors and non-aggressors (that aren't driven out). Clearly costs are being forced upon some of the freeloaders—energy of fleeing, disrupted feeding—as well as aggressors. As a reader, I would like to see a short clear possible scenario of the specific costs and benefits and their altruistic distribution, then a clear set up of the questions answered here and how they relate: Adjustments of flock size to support molt (which predicts what about repeated density assays?), changes or not to age-sex ratios if freeloaders are actually freeloaded?, etc. Similarly, to follow the arguments, a time-line of bunting arrival, aggression, moves between patches, etc. would be very useful in the Natural History section, where one could make clear what aspects are known, what are predicted under various cost-benefit scenarios, and what the authors assessed in this paper.

We have added a section in the introduction outlining the possible costs and benefits of aggression in flocking buntings in the context of altruist and freeloaders, as suggested by Reviewer 2. Immediately following this new section, we outline the three objectives of this study: (i) document patterns of aggression (ii) assess bunting densities and age ratios across habitat quality, (iii) evaluate possible proximate mechanisms underlying this aggression. However, we delay developing predictions of how flock composition and how the frequency of freeloaders and altruists should change according to levels of aggression until the discussion. Because the data in this paper cannot directly evaluate these predictions, we think these more specific predictions belong in the discussion.

Discussion—the basic argument that there is a freeloaded class of less-non aggressive individuals is clear. What is less clear is how one would discriminate the several ways in which birds could be acting as altruists, freeloaders, selfish or unselfish individuals, punishers, etc. I try to make this problem clear below and then note some specific problems with figures and tables.

Starting **P 14**. I think that the discussion of free-loading would be a lot clearer if the authors could put it directly in the current context—could operationalize the elements of theory with actual events. Who, behaviorally, are the freeloaders, the selfish punishers and how would this scenario translate to the bunting pre-molt context? Who are the unselfish altruists in this scenario—those who leave without being pushed out and without contesting access to the best foods? This is one of several places where the authors refrain from fully translating their situation into the two theoretical constructs within a multilevel selection framework. It would

greatly strengthen their argument if these translations were made clear—and the two scenarios do differ in how each would expect the bunting flock members to do cost-benefit-wise.

Agree. Throughout the discussion of the two multi-level selection models, we have clarified which sex/age class should be altruist, freeloaders, and/or “selfish punishers”. We agree that this creates stronger, more direct links between theory and our data/observations, and helps illustrate different predictions outlined by the two multilevel selection models.

P 16 | 377-379 introducing alternative non-adaptive or spandrel-style explanation: The authors should not call “behavioral syndromes” a necessarily non-adaptive explanation. A behavioral syndrome is simply a description of trait structure, that can, yes, potentially limit adaptive flexibility or response to selection IF selection bumps up against features of the syndrome that are not easily unlinked. But they don’t have to limit adaptation in a given situation. I didn’t find the discussion of this particularly appropriate to behaviors that may or may not be highly variable within an age-sex class. That would await further study of individuals and the question of whether particular individual phenotypes bear the brunt of costs, but (for instance) cannot disappear because those phenotypes were also the most successful breeders. *Agree. We clarified our wording so that behavioral syndromes and non-adaptive explanations are distinguished as TWO alternative hypotheses. We did not intend to lump behavioral syndromes as a non-adaptive explanation in our previous draft.*

P 16, L392—here in discussion the authors get less careful about attributing the regulation of aggression to DHEA, albeit saying “in part”. Perhaps regulating is too loaded a term.

Agree. We changed regulate to influence and toned down our wording of DHEA influencing aggression throughout the MS.

Tables 1, 2. I don’t follow what some of the variables are. What I “Minutes” measuring? Total time spent assessing chases? If so, the time periods are sometimes VERY short (1.87, 6.57) and vary all over the place. Minutes is used in calculating rate of chases. But if minutes is total time spent counting chases, that doesn’t seem adequate as basis for an estimate. Time of day? Number of days of assessment? For strong behavioral sampling, I would have tried to equalize sample effort.

We have clarified column definitions in Tables 1 and 2, and also added a new column that summarizes (i) the total time spent observing buntings and (ii) the number of focal buntings observed during each time period. We realize that this section was not intuitive, so we also clarified our methods to better reflect what these tables summarize.

P 25 Table 5 shows age-sex classes in September, but the year is 2007; I presume this is a typo because in the text 2006 was the year in which hormones were assayed.

Fixed.

Also it is interesting that the actual densities had not changed by the time molt arrived, in either year. (A drop between arrival densities and pre-molt densities in 2006 is explained by net placement.) Would one not expect a drop if chasing were effective in regulating incoming bird numbers?

Good point. Yes, theoretically it should as long as arriving buntings did not settle in areas where others had been displaced. Densities of green birds should be the most sensitive to changes as a result of aggression, but these changes could be muted by increases in adults to areas where green birds had been displaced from; we have clarified this in the results where Tables 1 and 2 (which summarize bunting densities before and during molt) are mentioned.

Table 4. There is something wrong here as the expected numbers (based on expected fractions of observations * total observations I assume) do not add up to the total observations (n=38), thus making the generated Obs-Exp incorrect. They do add up to same in Table 3 which uses the same approach.

Fixed.

Table 6. Need to put more info in legend—e.g. 50 x 50 cm plots, etc. Legends in general need to stand alone. Assuming that “mean differences” are what we are supposed to see from the means presented, shouldn’t there be SD or SE for these means?

Fixed. We added more details about how we gathered this data and added SE.

Fig 1. Needs a lot more info in legend to tell reader what these data are based on. It isn’t clear when the start of molt is, after arrival. (Is there a problem with age-ratio assessment in mid-late September if some of SY males have molted into AHY plumage, adding to AHY birds that have finished?)

Fixed. We have added to the legend that: these data are based on the first 20 individuals captured, that males molt, on average, 14 days earlier than females, but we cannot not provide specific dates as the timing of molt varies from year to year with the arrival of the monsoon rains.

We also clarify in the legend of Table 5 that timing of molt does not influence our assessment of age ratios as birds have not completed their molt by September. This was a very valid comment—thanks for pointing it out.

Fig 2. Why are the DHEA levels not separated by SY vs AHY birds? This shows an increase from arrival to pre-molt, while aggression levels at arrival in 2005 appear to be very high. And the overall DHEA levels should be very different between SY and AHY birds, if related closely to aggressive chases.

Fixed. DHEA is now separated by age and sex.

Appendix C

20 December 2019

Dear Editors of Royal Society Open Science,

Please find a revised draft of manuscript (RSOS-191510.R1): Despotism Aggression in Pre-molting Painted Buntings.

Review 2 rightfully pointed out that our discussion of behavioral syndromes was inappropriately cast as an evolutionary alternative to multi-level selection. We have clarified this section by explicitly stating that behavioral syndromes are not evolutionary explanations for the aggression we observed in Painted Buntings but could help explain variation in aggression within sex or age classes and, thus, influence an individual's decision to be an aggressive defender of food resources, a freeloader, or a selfish punisher.

(Please see page 16, lines 472-493, for the revised paragraph of behavioral syndromes).

This was a very good point made by Reviewer 2.

In addition, we have added all end statements (e.g., ethics, funding, data accessibility, etc.) to this final draft, following RSOS format. Please let us know if these require any changes.

We think this revised draft is logically robust and more clearly explained and look forward to seeing it in print!

Best,

Vanya Rohwer, Sievert Rohwer, John Wingfield